



# Reliability of Resilience Estimation based on Multi-Instrument Time Series

Taylor Smith[1], Ruxandra-Maria Zotta[2], Chris A. Boulton[3], Timothy M. Lenton[3], Wouter Dorigo[2], and Niklas Boers[3,4,5,6]

[1]Institute of Geosciences, Universität Potsdam, Germany
[2]Department of Geodesy and Geo-Information, Vienna University of Technology, Vienna, Austria
[3]Global Systems Institute, University of Exeter, Exeter, UK
[4]Earth System Modelling, School of Engineering & Design, Technical University of Munich, Germany
[5]Potsdam Institute for Climate Impact Research, Germany
[6]Department of Mathematics, University of Exeter, UK

**Correspondence:** Taylor Smith (tasmith@uni-potsdam.de)

**Abstract.**

Many widely-used observational data sets are comprised of several overlapping instrument records. While data inter-calibration techniques often yield continuous and reliable data for trend analysis, less attention is generally paid to maintaining higher-order statistics such as variance and autocorrelation. A growing body of work uses these metrics to quantify the stability

5   or resilience of a system under study, and potentially to anticipate an approaching critical transition in the system. Exploring the degree to which changes in resilience indicators such as the variance or autocorrelation can be attributed to non-stationary characteristics of the measurement process, rather than actual changes in the dynamical properties of the system, is important in this context. In this work we use both synthetic and empirical data to explore how changes in the noise structure of a data set are propagated into the commonly used resilience metrics lag-one autocorrelation and variance. We focus on examples from

10   remotely sensed vegetation indicators such as the Vegetation Optical Depth and the Normalized Difference Vegetation Index from different satellite sources. We find that varying satellite noise levels and data aggregation schemes can lead to biases in inferred resilience changes. These biases are typically more pronounced when resilience metrics are aggregated (for example, by land-cover type or region), whereas estimates for individual time series remain reliable at reasonable sensor noise levels. Our work provides guidelines for the treatment and aggregation of multi-instrument data in studies of critical transitions and

15   resilience.





## 1 Introduction

Observational records of climatic and environmental variables are not created equal – there exist large variations in the design, capabilities, and continuity of data sets. Many nominally continuous records are comprised of several different data sources which undergo design changes through time (Pinzon and Tucker, 2014; Moesinger et al., 2020; Gruber et al., 2019). While diverse records are generally tightly cross-calibrated, slight changes between different measurement periods have the potential to impact inferences based on those data – especially in indicators which are based on high-frequency changes in data structure. Cross-calibration procedures are also often geared towards maintaining means, long-term trends, or intra-annual consistency (Moesinger et al., 2020; Pinzon and Tucker, 2014), and may not properly address high-frequency data structures and higher statistical moments than the mean (Preimesberger et al., 2020).

It has been noted that measuring the mean state (i.e., the first statistical moment) of a system alone is not sufficiently informative for determining the dynamical stability (or resilience) of that system (Boulton et al., 2022). The stability of a system in question is tightly linked to the characteristics of its variability; the variance (i.e., the second statistical moment) and lag-one autocorrelation (AR1) have been proposed as stability indicators (Scheffer et al., 2009; Dakos et al., 2009; Lenton, 2011; Smith et al., 2022). Indeed, changes in these higher-order characteristics can reveal that a system is already committed to a stability gain or loss that would be impossible to infer from the mean state alone, and is not necessarily linked to observable long-term trends in the data (Scheffer et al., 2009; Boers, 2021).

However, variance and AR1 are potentially sensitive not only to shifts in the underlying processes associated with stability changes, but also to changes in equipment, measurement procedures or data processing schemes. Disentangling the effects of process and measurement changes can be very challenging in practice; assessing them quantitatively using synthetic data with ground reference known by construction can therefore help to understand whether observed changes in variance and AR1, for example, should be attributed to underlying changes in dynamical stability, or rather to non-stationarities induced by changing measurement characteristics.

In this study, we explore the impacts of non-stationary measurement characteristics on resilience estimates derived from different (dis-)continuous satellite records – the multi-sensor microwave Vegetation Optical Depth Climate Archive (VODCA, (Moesinger et al., 2020)), GIMMS3g AVHRR Normalized Difference Vegetation Index (NDVI, (Pinzon and Tucker, 2014)), and MODIS NDVI (Didan, 2015) – which have been used in several studies of vegetation resilience (Verbesselt et al., 2016; Boulton et al., 2022; Smith et al., 2022; Feng et al., 2021; Forzieri et al., 2022). The potential impacts of changing satellite mix – VODCA and GIMMS3g contain information from several different satellites and instruments – have not previously been considered in-depth with regards to estimating statistical resilience indicators such as variance or AR1. We first develop synthetic data which mimics the changing data structure of the VODCA and GIMMS3g data sets, and then explore resilience estimation at multiple levels of data aggregation, which serve as a proxy for global or regional aggregations of the spatial field of time series in question. We then compare the results of the synthetic experiments to data from VODCA, GIMMS3g, and MODIS NDVI to quantify the reliability of both local- and global-scale resilience estimates. We focus on satellite-derived





vegetation data, but our approach can in principle be adapted to discontinuous data records in general, from multi-proxy paleo-
climate reconstructions to simple instrument upgrades at long-term weather stations.

## 2 Methods and Data

### 2.1 Synthetic Data

The underlying structure of a satellite-derived vegetation time series can be thought of as having three parts: (1) the underlying
driving process given by vegetation growth and decline, (2) additional fluctuations including inter-annual variability and short-
term weather-driven effects that we refer to here as dynamical noise, and (3) additional noise due to imperfect measurement
or retrieval of the system, termed here as sensor or measurement noise. If we create a synthetic system where (1) and (2) are
static, we can observe the influence of varying measurement noise (3) throughout a given time series. We can further control
the instrument signal-to-noise ratio (SNR) by tuning the amplitude of measurement noise relative to the background signal.

We construct synthetic time series mimicking the structure of the VODCA data – that is, we generate daily data running
from 1987-2017, comprised of five satellite platforms (Moesinger et al., 2020) (Supplemental Table S1). We first generate a
time series $X(t)$ that represents a true underlying signal by integrating the following stochastic differential equation:

$$dX(t) = aX(t)dt + \sigma_{dyn}dW \tag{1}$$

with drift parameter $a < 0$, a Wiener process $W$ and dynamical noise amplitude $\sigma_{dyn}$, defining an Ornstein-Uhlenbeck process.
The dynamical noise term above, producing white Gaussian noise (dynamical noise), simulates background 'environmental'
variability (e.g., due to weather fluctuations).

Using this signal as a basis, we then add additional measurement noise according to the relative reliability of each sensor
that makes up the VODCA data set:

$$X_{satellite}(t) = X(t) + \sigma_{sensor}(t) * 1/R_{satellite} * 1/SNR \tag{2}$$

where the synthetic series $X_{satellite}$ is comprised of the true signal $X$ plus additional measurement noise $\sigma_{sensor}$ scaled by
the relative reliability of each satellite $R_{satellite}$ and a scaling factor $SNR$ to either increase or decrease the contribution of
measurement noise to the synthetic series. High SNR or reliability de-emphasizes measurement noise, low SNR or reliability
increases the contribution of measurement noise to the overall signal. The relative reliability of each satellite used in this study
can be found in Supplemental Table S1.

Finally, we mix the five sensors together by taking a daily mean, creating a single time series covering the whole time span
of the VODCA data. We then further aggregate this daily time series into a bi-weekly mean, to match the temporal resolution
of the NDVI data. We repeat this experiment 1000 times to generate a sample which can be used to examine the influence
of underlying changes in sensor noise across simulations. We create a further 100 iterations of this process (n=100x1000),



taking the median time series at each of the 100 iterations to assess the impacts of aggregating the underlying data or resilience estimates at different stages of analysis.

We perform a similar simulation procedure for GIMMS3g NDVI, which is comprised of a larger number of satellites (Supplemental Table S2, (Pinzon and Tucker, 2014)). However, we use a bi-weekly maximum value composite instead a bi-weekly mean, to better match the processing employed in the GIMMS3g product. Full code to reproduce our synthetic experiments can be found on Zenodo (Smith and Boers, 2022). Finally. we note that all correlations presented in this study refer to the Pearson's correlation coefficient.

## 2.2   Satellite Data

We rely on three satellite data records in this study: (1) Ku-band Vegetation Optical Depth (VOD) at 0.25° spatial resolution (daily, 1987-2017) (Moesinger et al., 2020); (2) GIMMS3g Normalized Difference Vegetation Index (NDVI, based on AVHRR) at 1/12° spatial resolution (bi-weekly, 1981-2015) (Pinzon and Tucker, 2014); (3) MODIS MOD13 NDVI at 0.05° (16-day, 2000-2022, (Didan, 2015)). To limit the influence of anthropogenic activity on our results, we mask out anthropogenic (e.g., 90  urban) and changing (e.g., forest to grassland) land covers using MODIS MCD12Q1 (500m, 2001-2017) land cover data for each data set. Finally, we remove areas with low long-term average NDVI (<0.1) to focus our analysis on vegetated areas.

    For both MODIS and GIMMS3g NDVI data, we remove cloud-cover and other artifacts using an upwards correction approach (Chen et al., 2004). We further resample VOD data to a bi-weekly time step to more closely match the temporal resolution of the NDVI data sets. Using these cleaned and consistently-sampled data, we de-season and de-trend the data via 95  seasonal trend decomposition by Loess (Cleveland et al., 1990; Smith and Bookhagen, 2018; Smith et al., 2022). Further details of the decomposition procedure, data correction, and land cover masking can be found in (Smith et al., 2022).

## 3   Results

### 3.1   Construction of Synthetic Data

Generally, changes in the amplitude of the measurement noise throughout a time series will – assuming that temporal correlations in the measurement noise decay rapidly – have opposing impacts on AR1 and variance; increasing noise will reduce autocorrelation while increasing variance. Hence, time-variable measurement noise will bias AR1 and variance towards anti-correlation, given no other changes to the system. We can test this with a synthetic experiment which roughly mimics the sensor input data of the VODCA product (Figure 1).

    As can be seen in Figure 1, the addition of multiple overlapping signals with different properties significantly changes the 105  dynamics of the signal through time. This effect, however, is also controlled by the overall SNR of the synthetic satellite data that are averaged together (Supplemental Figures S1-S3). When measurement SNRs are low, the underlying signal (Figure 1a) is lost; with higher SNRs, the effects of combining multiple signals are increasingly muted (Supplemental Figures S1-S3).





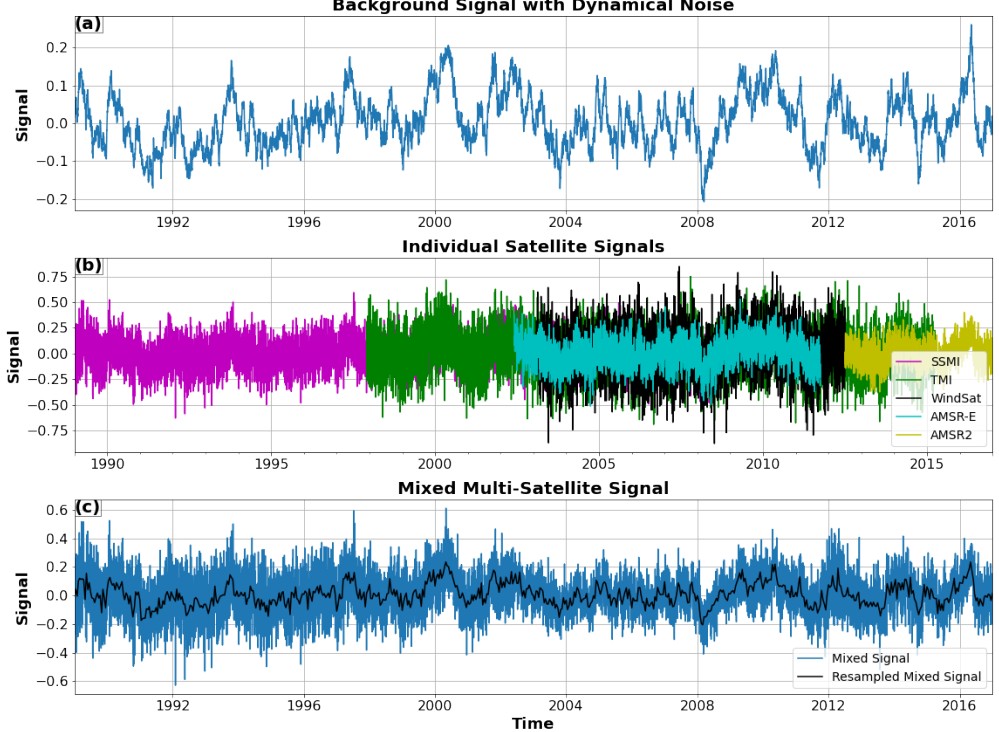

**Figure 1.** Synthetic experiment mimicking Vegetation Optical Depth (VOD) time series, with relative measurement noise scaling ($R_{satellite}$, see Methods) set to values between 1 for the most reliable sensor and 0.44 for the least reliable, and signal-to-noise ratio (SNR, see Methods) set to 1. (a) Ornstein-Uhlenbeck process with dynamical noise mimicking an underlying signal to be measured (see Methods). (b) Underlying signal plus additional white Gaussian measurement noise by individual synthetic sensor scaled by reliability $R_{satellite}$, based on the characteristics of the satellites used in the VODCA data set (Moesinger et al., 2020) (see Supplemental Table S1 and Methods for details). (c) Combined synthetic signal via taking the daily (blue) and bi-weekly (black) means.

## 3.2 Signal-to-Noise Ratios and Data Aggregation

The correlation of AR1 and variance for the raw underlying signal (i.e., for the synthetic Ornstein-Uhlenbeck process contain-
ing only dynamical noise, Figure 1a) is by construction high ($\sim$1), and is controlled primarily by the degree of autocorrelation in the underlying process (Supplemental Figure S4). In contrast, a signal with time-variable measurement noise will tend towards anti-correlation between AR1 and variance as the changing noise level pushes the two metrics in opposite directions (Figure 2). This effect can also be enhanced by aggregating multiple time series; changes in measurement noise that occur contemporaneously in all time series are emphasized. Constant underlying variation (dynamical noise) between time series is
removed to the extent that it is independent, which further increases the relative strength of the time-variable measurement noise in the aggregated signal.



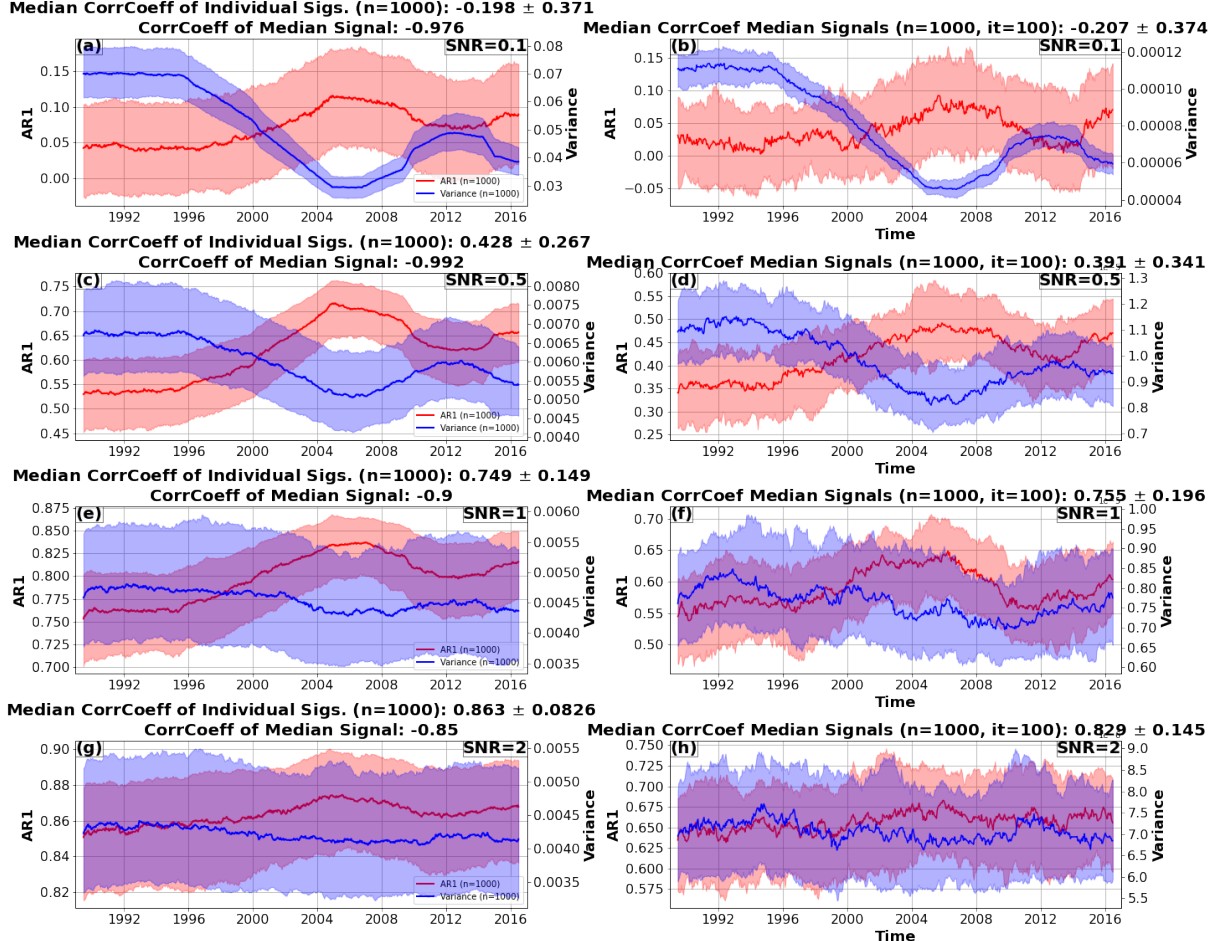

**Figure 2.** Effect of sensor signal-to-noise ratios (SNRs) and data aggregation scheme. Left column shows median (25th-75th percentiles shaded) AR1 (red) and variance (blue) time series of n=1000 synthetic time series. Right column shows the median AR1 (red) and variance (blue) time series of 100 iterations, taking the median time series from n=1000 synthetic time series each time before calculating AR1 and variance. The SNR increases from 0.1 (a,b) to 2 (g,h), while relative noise levels between satellites ($R_{satellite}$, see Methods) are held constant. Correlation coefficients ± one standard deviation listed in plot titles. Low SNRs produce anti-correlated individual AR1 and variance time series, indicating a bias induced by the changing sensor mix. For increasing SNRs, the correlation values between individual AR1 and variance time series become increasingly positive, indicating a weaker bias. The same holds true if the AR1 and variance of the medians of the time series themselves are considered (right column). However, correlations between median AR1 and variance time series remain negative, indicating that the bias persists in this case (left column). AR1 and variance are calculated on a five-year rolling window.

If we vary the SNR in our synthetic experiments we can examine to what degree changes in both the amplitude and structure of measurement noise are reflected in the resilience metrics AR1 and variance. We further compare three ways of calculating the correlation between AR1 and variance: (1) the median of the correlation coefficients of AR1 and variance of each individual





measured time series (n=1000), (2) the correlation coefficient of the median AR1 and variance time series (n=1000), and (3) the correlation coefficient of the AR1 and variance of the median synthetic time series (n=1000x100) (Figure 2). We note that we use the resampled bi-weekly means for our estimates of AR1 and variance (Figure 1c, black line); this does not impact our inferences from the data.

It is clear that changes in measurement noise – in the absence of changes to the underlying process – will lead to strong
anti-correlation in the two commonly used resilience metrics AR1 and variance (Figure 2). This effect is more pronounced for lower SNRs if the AR1 and variance of individual time series, or of median time series (Figure 2, right column), are considered. However, the correlation remains strongly negative when the median of large ensembles of AR1 and variance time series are considered (Figure 2, left column). Essentially, in this case the dynamical characteristics are averaged out, while the influence of the changing sensor mix, common to all time series, persists and dominates the AR1 and variance medians.

We emphasize that while the correlation between AR1 and variance is generally positive for *individual synthetic series* assuming reasonable SNRs, averaging their AR1 and variance time series leads to strong anti-correlation (Figure 2, left column). In contrast, first averaging the underlying signal and then computing AR1 and variance on that averaged time series leads to positive correlations at higher SNR (Figure 2, right column), indicating that the biasing effect of the sensor mix is attenuated. It is important to emphasize this point – averaging an ensemble of AR1 and variance time series leads to anti-correlation, and thus
to strong biases, while averaging the time series first and then calculating AR1 and variance leads to positive correlation, and thus to weaker biases; which features of the noise structure of the time series are emphasized by these two averaging schemes has a strong impact on the outcome, and hence the interpretation of changes in AR1 and variance through time.

### 3.3 Comparison to Global Satellite Data

It is possible to directly compare the results of our synthetic experiment (Figure 2) with similar global averages of pixel-
wise AR1 and variance estimates for both VODCA and GIMMS3g NDVI (Figure 3, Supplemental Figure S5). How well the changes in AR1 and variance for the synthetic and satellite cases match up is strongly controlled by estimates of the underlying measurement noise levels of each individual satellite record that comprises the VODCA and GIMMS3g NDVI data sets, respectively. Both satellite data sets exhibit anti-correlation between the global medians of AR1 and variance time series, to different degrees (Figure 3, Supplemental Figure S5). It is important to note that VODCA data are merged via cumulative
distribution matching and daily means (Moesinger et al., 2020), and GIMMS3g is aggregated by bi-weekly maxima (Pinzon and Tucker, 2014).

In the case of VODCA (Figure 3), where the relative noise levels of the individual satellite instruments is fairly well-constrained, the overall shapes of the medians of the AR1 and variance of the synthetic measured time series (red) match quite well to the global median AR1 and variance time series computed from VODCA (blue), especially at low SNR (Figure 3a,c).
At higher SNRs, the results for the synthetic data still generally follow the global pattern of the VODCA data (Figure 3b,d), but have a more positive AR1/variance correlation. This indicates that – when globally aggregated – AR1 and variance changes in the VODCA data set are to some degree controlled by changes in the underlying data structure. Inferences on actual stability or resilience changes, if based on large-scale averages or medians of AR1 and variance time series, are hence likely to be biased.





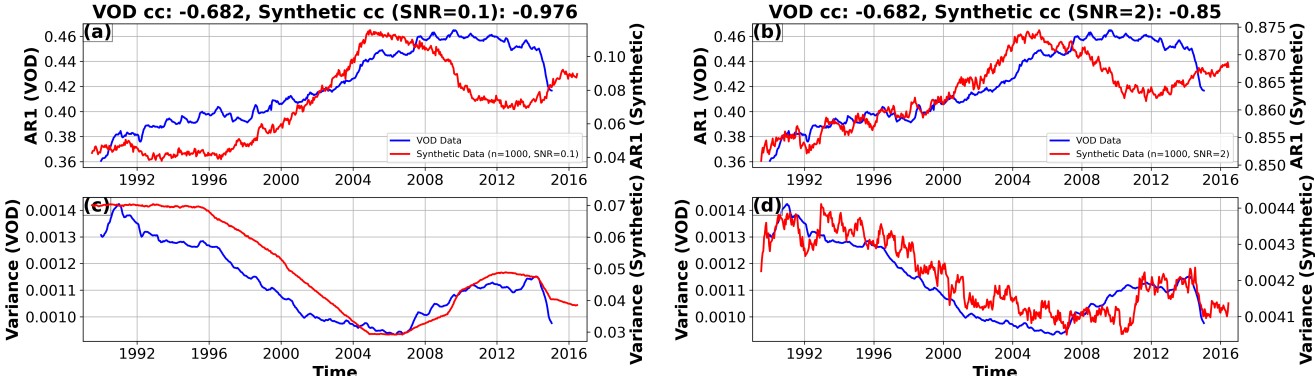

**Figure 3.** Comparison between real and synthetic data. (a,b) Median AR1 for synthetic data (red) and vegetation optical depth (VOD) data (blue, median taken over all AR1 series globally). (c,d) Same as (a,b) but for the variance. Left column shows low signal-to-noise ratio (SNR=0.1), right column shows SNR=2. AR1 and variance are calculated on a five-year rolling window. Correlation coefficients (cc) between AR1/variance are given in the individual panel titles. Both the synthetic and satellite data sets show negative correlation, modulated by SNR. Note that satellite and synthetic data are not plotted on identical y-scales. The global medians of AR1 and variance time series for the VOD data can be approximated to some extent by the corresponding medians of AR1 and variance time series for the synthetic data, which suggests that the global medians of AR1 and variance are biased by changes in the VOD data through time.

For the case of GIMMS3g NDVI, resilience metrics – particularly AR1 – do not match as closely to the synthetic data, especially in the case of low SNR data, where the influence of multiple overlapping sensors is strongly expressed (Supplemental Figure S5). As the GIMMS3g NDVI product uses a maximum value composite approach, the presence of many overlapping satellite data sets has a strong impact on AR1 and variance. It is also important to note that before 2000, GIMMS3g NDVI relies on AVHRR/2, and from 2000 onwards, the GIMMS3g NDVI relies on data from the AVHRR/3 satellite instrument (as well as SeaWiFS, SPOT, MODIS, PROBA V, and Suomi for calibration) (Pinzon and Tucker, 2014). This change in instrument sensitivity also likely contributes to changes in AR1 and variance through time (Supplemental Figure S5).

It is clear from both the synthetic (Figure 2) and real (Figure 3) data that aggregating many AR1 and variance time series leads to strong anti-correlation between AR1 and variance, suggesting biases caused by changes in the data structure and measurement noise. In the synthetic experiment – and at high SNRs – individual synthetic series maintain the expected positive correlation between AR1 and variance even when the median data exhibit anti-correlation (Figure 2, left column). As it is expected that real satellite data has a relatively high SNR (∼2+) (Salomonson et al., 1989), it is likely that individual time series also exhibit this positive correlation between AR1 and variance and that the biasing effect of changing sensor mixes will be weakly expressed. It is important to note, however, that not all land surfaces are equally well-measured – SNR can vary considerably based on instrument, spectra, and land-cover type.





### 3.4 Individual Time Series Correlations

It is important to distinguish between global-scale aggregations and the behaviour of individual pixels. If we perform our correlation analysis for the satellite vegetation data sets at the pixel-scale, we can explore the distribution of correlation coefficients between AR1 and variance at each grid cell and compare them to the corresponding distributions for the synthetic data for different SNRs (Figure 4). Correlation coefficients for satellite data are divided by land-cover type to reduce the number of points in each aggregation, and emphasize that the positive correlations between AR1 and variance are consistent across

multiple ecosystems.

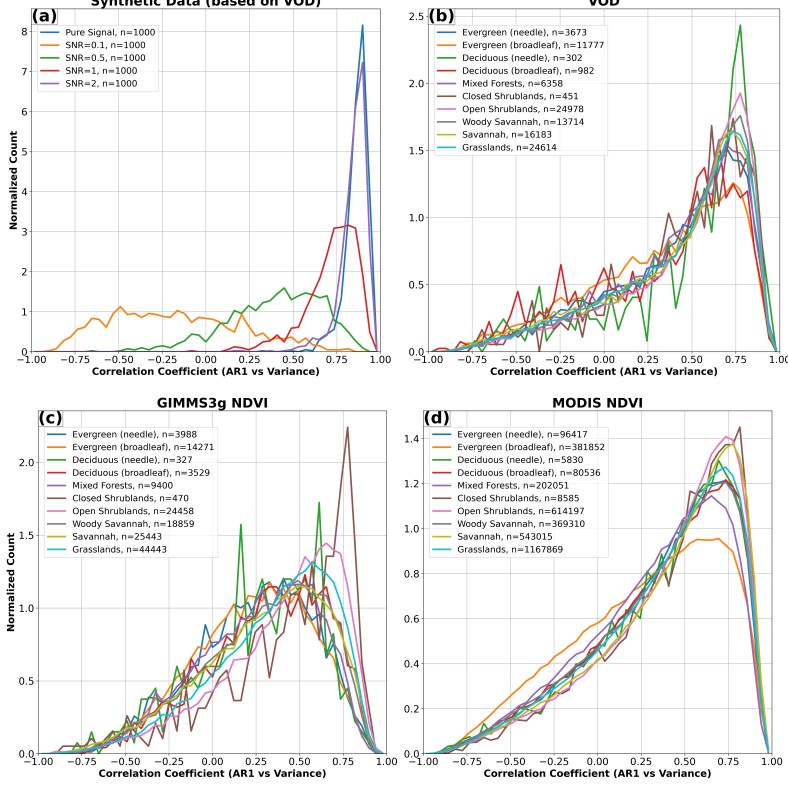

**Figure 4.** Distribution of correlation coefficients between AR1 and variance computed for individual time series. (a) Synthetic data based on the vegetation optical depth (VOD) data for different signal-to-noise ratios (SNRs), (b) VOD data, (c) GIMMS3g normalized difference vegetation index (NDVI), and (d) MODIS NDVI. AR1 and variance are calculated on a five-year rolling window pixel-wise (satellite data) and for n=1000 synthetic time series. Each panel has 50 equally-spaced bins from -1 to 1. For (a), four different SNRs are shown, as well as the underlying signal (i.e., without the influence of a changing measurement noise). For (b-d), pixels are divided by land-cover type. The distribution of correlation coefficients is generally positive, except in the synthetic approach with low SNRs.

As expected, the distribution of correlation coefficients between AR1 and variance is skewed towards negative values for synthetic data at low SNRs (Figure 4a). As there are no other processes in the synthetic data that would drive a change in



AR1 or variance, correlation is strongly influenced by changes in measurement noise through each individual time series. This, however, is not the case for the satellite data records (Figure 4b-d), or the synthetic data at higher SNRs (Figure 4a).

Both multi-instrument (VODCA, GIMMS3g NDVI) and single-instrument (MODIS NDVI) data show strong positive correlations between the individual AR1 and variance time series, across all land cover types. We posit that this is mainly due to changes in underlying processes driving vegetation through time – for example, inter-annual precipitation variability, long-term trends, or ecosystem changes. To lead to overall positive correlations between AR1 and variance, these changes would have to be larger than the shifts in noise driven by changes in the underlying satellite record. While we cannot rule out a residual

influence of those changes at the level of individual time series, these results suggest that individual-pixel AR1 and variance estimates are reliable; global- or regional-scale averages of AR1 and variance time series should be treated with more caution.

## 4   Discussion

Multi-instrument data is common across the environmental sciences. While long-term records generally aim to create continuous and tightly cross-calibrated data, they do not always maintain continuous higher-order statistics, such as the variance

and autocorrelation structure of the data record (Pinzon and Tucker, 2014; Moesinger et al., 2020; Markham and Helder, 2012; Claverie et al., 2018; Smith et al., 2008). This is partially by design – there are vastly more studies examining mean states and long-term trends in data than those focused on higher-order statistics, and especially AR1 and variance as resilience indicators. Nevertheless, an increasing number of studies have investigated resilience shifts based on these indicators using multi-instrument data.

We focus here on two discontinuous data sets – VODCA and GIMMS3g – which have been used in recent investigations into the resilience of vegetation regionally and globally (Smith et al., 2022; Boulton et al., 2022; Feng et al., 2021; Rogers et al., 2018; Hu et al., 2018; Wu and Liang, 2020; Jiang et al., 2021). The validity of these multi-sensor records for analyzing vegetation resilience has been debated (Smith et al., 2022); indeed, we find that much of the global-scale structure of VOD resilience changes – if inferred based on large-scale aggregates of AR1 and variance time series – can be reproduced with

synthetic data mimicking a changing mix of sensors (Figure 3). While synthetic data does not match as closely for AVHRR (Supplemental Figure S5), questions remain about the degree of influence changing data structure has on interpretations of changing resilience.

    Based on our results, we infer that the correlation between AR1 and variance time series can serve as a rough proxy for the strength of the biases caused by combining different sensors; the more negative (positive) the correlations, the stronger (weaker)

this effect will be. While the residual influences of changing measurement noise cannot be strictly ruled out – especially in a real-world case where measurement noise is influenced by a wide range of factors – individual AR1 and variance series exhibiting positive correlations indicate that process-level changes, rather than measurement noise, dominate AR1 and variance signals. Our analysis of synthetic data (Figure 2, Figure 4) shows that reasonably high SNRs strongly reduce the influence of changing measurement noise on resilience metrics at the individual time-series level or if resilience metrics are computed from

aggregated time series of the system state, even given relatively large variations in measurement noise through time. However,





our results also show that taking large-scale averages of AR1 and variance time series – especially over incoherent spatial regions – should generally be avoided as this tends to amplify the biases induced by changing sensors.

Our findings have important implications for recent regional- and global-scale analyses of vegetation resilience based on VODCA (Smith et al., 2022; Boulton et al., 2022) and GIMMS3g (Feng et al., 2021). All three papers rely primarily on

individual-pixel level analyses to support their inferences about changing resilience patterns, which our results indicate are reliable (Figure 2, Figure 4); indeed the vast majority of VOD and GIMMS3g time series globally have positive AR1 and variance correlations (Figure 4). However, these three papers also present spatially aggregated trends in resilience indicators – for example, Extended Data Figure 6 in Smith et al. (2022) and Figure 2c in Boulton et al. (2022) – that should be treated with caution, as our synthetic experiments indicate that aggregated AR1 and variance time series can be strongly influenced by

changes in measurement noise (Figure 2).

Despite the strong positive correlations seen between AR1 and variance at the individual-pixel level (Figure 4), changes in satellite mix will still influence long-term estimates of resilience, especially if AR1 and variance are aggregated across multiple time series. This impact is unfortunately difficult to quantify – the underlying noise of a satellite data record is highly sensitive to the individual characteristics of the location being monitored, and measurement noise can change drastically in

time and space. Coastal areas with strong atmospheric moisture signals will behave differently than dry continental interiors; these differences will be sensitive to diverse factors (e.g., sensing wavelength, time of day of overpass, satellite footprint size). Thus, not all time series are equally reliable, and the influence of sensor noise on resilience metrics could vary widely. Without a strong handle on the underlying driving process, any quantification of changing satellite noise will be difficult to disentangle from changes in the ecosystem being measured.

There is thus no safe and efficient way to correct for this influence globally; simple strategies such as removing the global average signal from each individual time series as a normalization procedure create the risk of destroying any underlying changes in AR1 and variance that are in fact due to changing resilience, such as those potentially driven by global or regional environmental changes. If, however, there is a strong reason to believe that a region will behave coherently, some of the influence of changes in satellite instruments can be removed via first aggregating time series from such a region and then

calculating AR1 and variance on the resulting mean time series (cf. Figure 2, right column). Conversely, first calculating AR1 and variance and then averaging those metrics over a region will emphasize changes in the data structure unrelated to resilience changes (cf. Figure 2, left column). It is thus important to consider carefully at what stage data is aggregated, and how to interpret regional- or global-scale changes in AR1 and variance.

## 5   Conclusions

Our analysis highlights the potential pitfalls of using multi-instrument or discontinuous data to monitor the commonly used resilience indicators given by lag-one autocorrelation and variance. We find that when time series of these resilience indicators are aggregated – as it would typically be done to show regional- or global-scale changes – the influence of changes in the underlying data structure is enhanced, leading to potentially erroneous and biased interpretations. On the other hand, both





synthetic and empirical experiments indicate that – given reasonable signal-to-noise ratios – process-based or environmental
changes in individual time series are a more important driver of changes in resilience indicators than changes in the measure-
ment process. This is an important insight that emphasizes how best to aggregate, present, and interpret changes in resilience
across disciplines. We emphasize that single-sensor instrument records – when available – should be preferred for analyses of
system resilience.

*Code and data availability.*  Data used in this study is publicly available (Moesinger et al., 2020; Pinzon and Tucker, 2014; Friedl and Sulla-
Menashe, 2015; Didan, 2015). Code to reproduce the synthetic data used in this study can be found on Zenodo (Smith and Boers, 2022)

*Author contributions.*  T.S and N.B conceived and designed the study and interpreted the results. T.S. processed the data and performed the
numerical analysis. T.S wrote the manuscript with contributions from all co-authors.

*Competing interests.*  The authors declare no competing financial interests.

*Acknowledgements.*  The State of Brandenburg (Germany) through the Ministry of Science and Education and the NEXUS project supported
T.S. for part of this study. T.S also acknowledges support from the BMBF ORYCS project, and the Universität Potsdam Remote Sensing
computational cluster. N.B. acknowledges funding from the Volkswagen Stiftung, the European Union's Horizon 2020 research and innova-
tion programme under grant agreement No. 820970 and under the Marie Sklodowska-Curie grant agreement No. 956170, as well as from the
Federal Ministry of Education and Research under grant No. 01LS2001A. This is TiPES contribution #X.





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
