# Peer review of "Reliability of Resilience Estimation based on Multi-Instrument Time Series"

_Earth System Dynamics, 2022_

## Author Comment (AC1)

**Referee #1**

*This manuscript addressed an interesting topic by investigating whether measurement noises can impact the inference of resilience using remote sensing data. They used a simulation approach to investigate how signal-to-noise ratio (SNR) influences the calculations of two indicators of resilience, namely lag-1 autocorrelation (AR1) and variance. Their results have implications for assessing the possible impact of measurement errors in observational data. Overall I found this study well designed and conducted. I have a few comments that may help improve the paper.*

Thank you for your time with the manuscript and the helpful comments! We will address each comment individually below.

*(1) the study generated simulated time series by combining a background time series with a number of random noises. I was wondering if this characterized the realistic errors introduced by changing instruments. I was not an expert in remote sensing, but I thought in some cases the change of instrument might induce sudden increase/decrease in the time series (rather than a small noise term). Such changes can have major impacts on the calculation of resilience indicators, keeping in mind that such indicators were used to detect 'sudden changes' in the time series, whether they were due to measurement errors or underlying processes?*

Thank you for this comment. Sharp jumps are indeed a potential issue in time series obtained from combining signals from different sensors that are active over different time periods. There are many different studies on how and when to cross-calibrate long-term data to handle this. For example, in the VODCA data set that we attempt to model, biases between sensor absolute values are rectified with a CDF matching approach.

Given that problems arising from mean-shifts can be handled comparably easily, in our synthetic set up we address problems that go beyond a mean-shift. We assume that any composited multi-satellite data has had such shifts removed; our analysis rather focuses on what is *not* well treated in such common data compositing/calibration schemes – namely changes in higher-order statistics such as variance or autocorrelation.

*(2) AR1 and variance are two important indicators of resilience, or early warning signals (EWS) for catastrophic changes, but there are more. Moreover, researchers had been developing composite EWS by combining different metrics. Given that measurement errors may influence AR1 and variance differently or in opposite directions, I was wondering if a composite EWS would be more robust to measurement errors.*

You are right that some studies have proposed combinations of the Variance and AR1 coefficient, as well as other indicators such as Skewness or Kurtosis. In the present case, we focused on the Variance and Lag-1 autocorrelation as the underlying theory provides clear equations relating them to the recovery rate $\lambda$ and – in this sense – to the resilience of the underlying system: $\langle x^2 \rangle = \sigma^2/(2\lambda)$ and $\alpha(1) = \exp(-\lambda\Delta t)$. We believe that in order to obtain a detailed understanding of the effects of changing sensors on these indicators, it is best to keep them separated and not combine them, at least in the context of our study. Formulating a combined resilience indicator would make it more difficult to attribute the individual effects of the changing sensor mix to changes in system parameters (e.g., system memory – autocorrelation, system variability – variance).

*(3) the authors discussed about the difference between the average of variance from individual time series and AR1 of the aggregate time series, particularly their different behaviors in the presence of measurement errors. Similarly, the reference Feng et al. (2021) found different temporal trends of these two metrics. However, these two metrics represent different properties (i.e., local- vs. larger-scale resilience) and they did not necessarily exhibit different patterns, even if there was no measurement error. The problem is, the local-scale variance did not add up to give the larger-scale variance, but modulated by the synchrony between local grids. I attached a theoretical paper illustrating this:*

*Wang, S. & Loreau, M. Ecosystem stability in space: α, β and γ variability. Ecol. Lett. 17, 891–901 (2014).*

Thank you for providing that reference – this is a very interesting piece of work! This is something that we did not really consider in our work: how does the variance of individual time series match up to the aggregate variance of many time series in the absence of problems due to combining data from different sensors. Our focus was rather on how resilience analysis has typically been presented (e.g., with regional mean time series), and what problems there might be with that approach given data with dynamic uncertainties. It is not unexpected, however, that the sum of many individual variances does not add up to a single mean-series variance – these two cases would be exposed to quite different noise levels (e.g., random fluctuations are suppressed in a multi-series mean). While we find this a very interesting problem for further exploration, we feel that multi-scale and spatially-conditioned changes in resilience are outside of the scope of this current work. We will, however, add an explanation in the revised discussion that differences between the variance/AR1 of aggregated time series and aggregated variance/AR1 time series may also be due to differences between the local- and larger-scale resilience.

*(4) while the manuscript was overall well written, I had to say that I was confused by the different metrics involved in the figures, which seemed to be quite related but differ in important ways. For instance, the authors calculate resilience indicators using several approaches, e.g., deriving the numbers for an individual time series, first aggregating the time series and then calculating AR1 and variance, or first calculating AR1 and variance and then averaging them. They also calculate correlation between AR1 and variance at different levels of complexity. I would suggest to add a table to clearly define all key metrics in the figures, with explanations what a positive/negative or higher/lower value mean.*

Thank you for this suggestion – we endeavored to keep our terminology as clear and explicit as possible, but it remains difficult to parse in some places! Adding a comprehensive table is an excellent suggestion, and we will do so if a revision of the paper is requested.

*Specific comments:*

*L52: What does 'synthetic series' mean? I think it is simply a simulated time series.*

Yes, we refer here to simulated time series. To keep our language specific, we refer to each individual series as 'synthetic', and when we produce multiple realizations each is referred to as a single 'simulation'. We will make sure this is clear in a revised version of the MS.

*L79: How was this 'aggregating' implemented?*

In each case of aggregation, we use either a time-explicit (e.g., daily) mean, or a time-explicit maximum value. To simulate VOD, we aggregate multiple instruments with a mean. To simulate AVHRR, we take the multi-week maximum, to match the method used in the original data set. For mixing multiple multi-instrument series, we use a time-explicit mean (e.g., averaging 100 synthetic series into a single averaged series). We will clarify this in a revision.

*Figure 2: Not sure that I understood these figures correctly. Did the 'median corrcoef median signals' on the right represent the median of the 'corrcoeff of median signal' on the left? Why they were so difficult, even by sign?*

This is admittedly a rather dense figure. We think that the suggestion of the referee to add a table will greatly help in describing the differences in the metrics and making this figure more accessible. On the left, AR1/Variance are first calculated, and the median of 1000 iterations is displayed. On the right, we take the median signal from 1000 simulations, then calculate AR1/Variance on that. We perform that calculation 100 times, and display the median of that result on the right panels. The difference between the columns comes from when the data was aggregated – aggregating many time series and then calculating AR1/Variance produces a significantly different result from calculating many AR1/Variance series and then aggregating those.

*L130: "the correlation between AR1 and variance is generally positive for individual synthetic series" – any result supporting this argument?*

This refers to the labels of the left hand column (Fig 2) and the histogram shown in Figure 4 – for higher SNRs, the median correlation coefficient between individual-series AR1/Variance is positive, while it is negative when those AR1/Variance series are instead aggregated. We will update the reference on this statement to be clearer.

*Figure 3: I must miss something. How to determine the SNR in the real data? And did it happen to exhibit SNR = 0.1 and 2 in the empirical data?*

Determining the SNR of the real data is indeed challenging – there are many factors that influence sensor noise, and most change dramatically in space and time even for individual sensors. Cloud cover, atmospheric water content, and time-of-day of satellite overpass are a few key influences on local-scale sensor noise. Since we cannot quantify sensor noise effectively in space and time, we instead aimed to construct a synthetic experiment which mimicked the global-scale patterns seen in VOD (e.g., blue lines) using only noise-level changes (red lines). SNR is then a secondary factor – this allows us to vary not only the relative noise between sensors (e.g. through time) but also the amplitude of that noise relative to the underlying signal. The values of 0.1 and 2 are thus not related to the SNR of the real data, but rather constraints we placed on the synthetic data to illustrate the potential influence of not only changes in noise through time, but also how the amplitude of that noise plays a role. Note that we deliberately chose very low values for the SNRs; in reality, a sensor with SNR equal to 0.1 will of course not be considered useful, and even SNR = 2 is still far lower than what is reported for operational sensors. MODIS aims for SNRs of individual bands between ~70-1000

(https://modis.gsfc.nasa.gov/about/specifications.php); Landsat 8 OLI aims for SNRs of at least 100 (https://landsat.gsfc.nasa.gov/satellites/landsat-8/spacecraft-instruments/operational-land-imager/oli-requirements/); Sentinel-2 aims for SNRs around ~100 (https://sentinels.copernicus.eu/web/sentinel/user-guides/sentinel-2-msi/resolutions/radiometric). Our values of SNR from 0.1-2 are rather related to our synthetic experiment than trying to match real-world data.

*L242: what does it mean by 'aggregated'? You had explained that first aggregating time series and then calculating AR1 and variance can remove the influence of changes in satellite instruments to some extent. So did you mean 'first calculating AR1 and variance and then taking the averaging of these metrics'?*

Yes, in this case we mean that when many AR1/Variance series are averaged, changes in the noise structure through time are emphasized. Conversely, first averaging many series and then calculating AR1/Variance on the resultant averaged series reduces the impact of changes in sensor noise through time. We will clarify the use of the term "aggregation" throughout the manuscript in a revision.

---

## Author Comment (AC2)

*The impact of data noise on estimating two resilience metrics, variance and lag-1 autocorrelation, with satellite data was assessed in this study. The topic addressed is very importance, because satellite products are widely used to quantify the resilience of terrestrial ecosystems. My major concern is that it is within our expectation that data noise will affect the reliability of the metrics, what's the new finding of this study? I hope two aspects may be investigated in depth: 1. What's the uncertainty of the existing satellite products when used for quantifying resilience? For this purpose, the 'real noise' of the data needs to be quantified. 2. What's the uncertainty of using the products to depict the temporal changes in ecosystem resilience? For this purpose, the temporal changes in the noise are also need to be quantified.*

Thank you for your comments and the time spent with the Manuscript. We will address your three main points individually.

*1. What's the new finding of this study?*

As you note, satellite products are widely used to quantify the stability and resilience of different natural systems. These estimates of stability changes are based on the assumption that changes to higher-order statistics (e.g. variance, autocorrelation) are due to changes in the system under observation, and not in the observation mechanism. However, this assumption is not always true for satellite data – many nominally continuous data sets are in fact made up of a constellation of sensors. You are clearly right that it is within our expectation that data noise will affect the reliability of resilience metrics. As satellite data is used more and more often in these contexts, however, we felt it was important to explore how exactly changes in measurement procedures could propagate into resilience estimates, and what strategies might be used to minimize this issue. This is why we set up a thorough investigation of the detailed effects of combining signals from different sensors, using synthetic time series constructed for a wide range of possible signal-to-noise ratios. We further compared our controlled synthetic experiment to three real-world data sets, showing that while individual time series might be reliable, averaging individual time series of resilience indicators over large regions will tend to enhance the effects of measurement changes and thus reduce the reliability of resilience estimates. We feel that this is an important contribution to best practices when analyzing resilience based on satellite data. We will clarify the above points in a revised manuscript.

*2. What's the uncertainty of the existing satellite products when used for quantifying resilience? What's the uncertainty of using the products to depict the temporal changes in ecosystem resilience?*

Answering this question is exactly the reason we conducted our investigation. We undertook our study using synthetic data because we cannot fully constrain real-world data – satellite data noise shifts drastically in space and time, and we have no 'perfect' measurement against which to compare as a ground truth. Satellite instruments have reported signal-to-noise ratios and are calibrated against known quantities (e.g., deep space, the Sahara); however, these do not account for all sources of noise in, for example, a vegetation measurement. Other factors – such as atmospheric water content, cloud cover, and satellite viewing angles – will also influence estimated surface parameters. Without a true reference, dis-entangling the time-variable (e.g., seasonal, annual) changes in noise from changes in system stability is not possible with real-world data. Hence, we focused here on a controlled

synthetic system to explore the potential influences of changes in measurement procedures through time.

The lack of true reference also limits our ability to quantify the uncertainty in changes in resilience in real-world applications. We instead attempted to mimic a change in resilience proxies (autocorrelation and variance) using only changes in measurement noise; this could be thought of as a null model for whether or not there was a change in resilience. If changes in autocorrelation and variance are stronger than those implied by the changes in the underlying noise, than that could be interpreted as a more robust resilience change signal. Again, however, in a real-world system we would need a time-explicit estimate of how signal-to-noise ratios are changing to control for the influence of measurement procedure on our resilience estimates.

3. *the 'real noise' of the data needs to be quantified … the temporal changes in the noise also need to be quantified*

While we fully agree with your desire for time-explicit noise models for different satellite data, that is not the problem we set out to address with this publication. We instead approached the problem from a synthetic perspective, where we could control the amount and timing of noise, as well as how and when different sensors were mixed. Our work emphasizes the need for better constraints on the noise levels of satellite products and demonstrated the influence of varying signal-to-noise ratios on different ways to estimate resilience. It does not aim to provide a thorough accounting for different remote sensing products. We will revise our introduction to make this clearer.

---

## Author Response (AR2)

**Reviewer 1**

I appreciate the authors' careful revision and response to my comments. I am satisfied with most changes/explanations, but I would suggest the authors to expand the discussion on the practical implications of their results. In particular, in light of the results here, how people should analyze resilience using remote sensing data? Or they should stop such kind of analyses, e.g., in certain context? To this end, I still feel that the robustness of composite EWS to SNR should be evaluated. I understand that the theoretical link between EWS and SNR is less clear than the individual metric of EWS. But I believe that the objective of the current study is not to illustrate theoretical relationships, but instead assessing the robustness of resilience metrics in practical uses.

Thank you for your comments, and for taking time to re-review the Manuscript.

The main practical outcome of our work is the insight that multi-satellite (or any multi-sensor data, e.g., paleo-climate data from multiple proxies) has certain potential limitations that need to be taken into consideration when examining system resilience based on such data. Namely, averaging resilience indicators from a large number of different observational time series can serve to enhance measurement issues that aren't dominant in the dynamics of individual time series. Given reasonable signal-to-noise ratios (i.e., those found on modern satellite platforms), individual pixel (location) time series are still likely to yield reliable resilience estimates. We provide a rough proxy for that reliability – the correlation between variance and autocorrelation through time. The more positive this correlation, the smaller the unwanted influence from combining different sensors and vice versa. Indeed, we show that the correlation increases as signal-to-noise ratios improve (Figure 4).

What should hence in general not be done is averaging large ensembles of resilience estimates based on multi-satellite data, as that will serve to enhance spurious signals from changing instrumentation. This is a key insight that can be applied in future studies which look at large-scale, regional, or global aggregations of resilience estimates through time.

Composite resilience metrics preclude the possibility of comparing metrics that should (but don't always) agree. In particular, the theory of critical slowing down demands that individual changes in variance and lag-1 autocorrelation have to be consistent in order to be interpretable in terms of resilience changes; by combining them, it would not be possible to check this anymore. Indeed, we strongly focus on comparing the behavior of variance and lag-1 autocorrelation (via correlation coefficients) in our paper to tell apart actual resilience changes from spurious influences stemming from combining different sensors.

We have updated our Discussion to include an additional explicit note on the practical implications of our results.

Smith et al. investigated the reliability of resilience metric with a focus on the non-stationary characteristics of the measurement process. This arouses rethinking of the reliability of resilience conclusions derived from satellite-based vegetation products in current studies. I agree with the point that higher-order moment should be considered in resilience/sensitivity studies.

Thank you very much for taking the time to review our Manuscript. Addressing your comments has helped us further improve the presentation of our results. We will respond to your individual points below, together with references of what we changed in accordance. For clarity, we have broken some of your paragraphs into individual questions to make our response easier to follow.

However, I have several questions for the construction of the synthetic datasets as described by the authors. Also, I strongly recommend the authors to add associated backgrounds/descriptions/explanations in the introduction/methods sections, e.g., why the authors calculate correlation between AR1 and variance suddenly?

The theory of Critical Slowing Down, upon which we base our analysis, stipulates that variance and autocorrelation should be correlated through time, as they follow and respond to the same underlying process (see e.g. Boers, 2021 or Smith et al., 2022, and references therein). If the two variables are not correlated, the theory does not hold – we cannot say whether a state transition is approaching or more generally, if the system resilience is declining. On the other hand, increasing measurement noise leads to increasing variance but at the same time to decreasing lag-1 autocorrelation and vice versa. Hence, negative correlations between AC1 and variance suggest that their changes are caused by measurement / sensor issues rather than resilience changes and call for caution when using such data for resilience estimation. This has been clarified in the Introduction in Lines 31-39.

Why we should care about the aggregation process?

Aggregation destroys fine-scale spatial/temporal variability that is used to monitor changes in the higher-order signal dynamics (e.g., AC1, variance). Different aggregation processes will have different impacts, depending on how exactly the aggregation is performed. In general, aggregation will suppress the fine-scale variability that we are interested in for resilience estimation. This is mentioned in Line 36.

What the aggregation means, temporally or spatially?

In this paper, we use synthetic surrogates (e.g., 1000 simulations, cf. Figures 2, 4) to represent a spatial field of data. This is an idealized simulation – we don't, for example, model spatial autocorrelation between simulations – to focus on the influence of aggregating multiple data sets with similar measurement properties. We then examine how that aggregation serves to enhance or suppress different signals in our resilience proxies AC1 and variance.

Temporal averaging is performed in a few overlapping ways aimed at mimicking the construction of multi-sensor satellite vegetation data. We start in all cases with daily data from sensors with overlapping time periods. We then average these multiple satellite time series to daily temporal resolution, and then again to a bi-weekly median (for VOD) or to a bi-weekly maximum (for AVHRR) to follow how this data has been pre-processed in previous publications (Pinzon and Tucker, 2014; Smith et al., 2022; Boulton et

al., 2022). We do not aim here to determine the 'best' means of data fusion, but rather to highlight how such data aggregation in general can introduce spurious signals to resilience estimates.

The authors did not show related studies and I got lost when I read here. Please make sure interdisciplinary readers can also understand what you want to convey.

The issue of multi-satellite data and resilience estimation has not really been discussed in the literature so far – this is the gap that our paper aims to address (Lines 50-52). While some studies – for example, the publication presenting the VODCA data used here (Moesinger et al. 2020) – consider the autocorrelation of the data as a reliability metric, they do not explicitly investigate the impacts of data fusion on the signals used for resilience estimation. This is a key gap in the literature, as several studies have used multi-sensor satellite data (Feng et al., 2021; Smith et al. 2022; Boulton et al., 2022) without fully considering the implications of data fusion. We have clarified our introduction to add more context to this discussion.

In addition, the aggregation issue pointed out by the authors are a common sense in the remote sensing and GIS discipline. For a pixel (a mixed pixel) featured with strong spatial heterogeneity, using one value to represent the resilience for the whole area may lead to biased conclusions. I do not understand why the authors try to aggregate the resilience values since they already have resilience values with a high spatial resolution. Maybe the authors just want to reveal something using synthetic data. However, in practice, I would not "first calculating AR1 and variance and then averaging those metrics over a region". That makes no sense. The authors may add several studies that did this process to support your experiment design. Otherwise, will this conclusion benefit other studies, e.g., resilience estimates, in practical?

Many studies which have examined vegetation resilience over large areas present data not only in map view (e.g., one value for a pixel), but also in chart view (e.g., one value per time step). It is this secondary case where issues arise, and this has certainly been done repeatedly in the literature (e.g., Forzieri et al., 2022; Boulton et al., 2022; Smith et al., 2022; Feng et al., 2021 and references therein). Such aggregations are commonly used to illustrate differences between regions (e.g., changes in resilience in North vs South America, Amazon vs Congo), time periods (e.g., Amazon changes in the 90s vs 2000s), or land cover types (e.g., Savanna vs Forest changes and their relative strengths over time). Our results indicate that these spatial aggregations – when done by first calculating resilience estimates and then averaging – have the potential to introduce biases (cf. Figure 3 of the MS). On the other hand, first averaging the raw data (e.g., create a single time series for the whole Amazon), and then calculating changes in resilience through time, has a lower potential to introduce bias, with the important caveat that such averaging removes a lot of actual variability that might be relevant for resilience estimates. Moreover, the averaging should only be done over similar pixel time series. That is, don't aggregate forest and savanna together into a single time series, as this would create mixed-pixel signals which are difficult to interpret and might lead to strong biases. Understanding how averaging different signals can lead to biases is the main motivation for our study, and is applicable to any study that deals with sets of time series based on multi-instrument measurements. We have clarified the point of averaging our synthetic data in the Introduction (Lines 53-55). Several papers which have followed similar steps are also discussed in the Discussion (Lines 231-238).

The authors recommend to use single instrument record. Could I interpret this point of view as current data fusion studies are not reliable or not necessary? I found the authors did not clearly show their setups for the synthetic data, e.g., what is the justice to setup the error variance value? Arbitrary or have referred to associated references?

We do not think data fusion studies are irrelevant – indeed, constantly improving data fusion methods may help ameliorate some of the issues we found here. However, most data fusion studies are geared towards maintaining a continuous mean state or preserving underlying trends – not maintaining higher-order statistics, which would be crucial for estimating resilience reliably. Hence, resilience studies – which rely on those higher order statistics – should use caution when assessing multi-instrument data. If single-instrument data is available, this would eliminate one possible source of bias in interpretations of changing system resilience.

The chosen data noise values in our synthetic time series are arbitrary, and are used to make the point that signals that might be interpreted as a change in resilience (e.g., Figure 3) can be reproduced by time-variable measurement noise (see Lines 114-115). We could vary these noise levels to make the same point about anti-correlated variance/AC1, but would not be able to visually/qualitatively show how well our synthetic experiment can reproduce the findings of a real data set (i.e., Figure 3).

Will different value change the results and conclusions?

Different noise values for the synthetic data would not change our results or conclusions; however, the synthetic data would no longer match up as well to the VOD/AVHRR global composites we present (Figure 3). As we want to show the practical value of our results, we chose noise levels that allowed us to visually present our findings about time-variable noise and potential biases in resilience estimates. We also show a broad range of signal-to-noise ratios to make our results general; also note that these biases are smaller for high-fidelity data, but do not disappear.

Data quality control and integration methods (rescaling technique, weighted average method etc.) are important issues in data fusion studies […]

We fully agree that data fusion studies are reliable in many contexts, but are not necessarily geared towards preserving higher-order statistics as needed for the resilience measures we focus on. The amount of noise removed by various data fusion techniques is not necessarily the issue here, but rather that the *relative amount of measurement noise* can change throughout the time series when multiple overlapping instruments are used. That is exactly the change that might be erroneously interpreted as a change in system resilience, but is rather due to changes in data amount, quality, or sampling through time. This paper aims to illustrate this potential issue, and explore ways to mitigate the problem.

[…] and the fusion results typically have better quality than single satellite-based product (reduced random errors). I did not find new things here.

In general, if signals from different sensors are combined, we have more reliable data by removing random errors. However, the fusion needs to guarantee stationarity in higher-order statistics, which is not common practice; mean-adjustments are much more common. Thus, common data-fusion methods in practice remove or bias exactly those signals required for resilience estimation, while improving signals needed for investigations of the mean-state or mean-shifts (which are not our focus here); reduced random errors will suppress variance and increase autocorrelation during times where there

are more overlapping satellites without changing the underlying signal – this is exactly the kind of change that might be erroneously interpreted as a change in system resilience (Lines 34-38). We do not believe that data fusion is unnecessary or somehow problematic – it is an extremely useful technique for many contexts. However, fusion techniques that are optimal for some contexts can introduce biases in different settings.

Specific comments: [L. denotes Line]

L. 74 The authors integrated the five sensors by taking an average of their values. That is, the authors regarded the weight given to each sensor is equal. However, given that the authors have defined/setup the relative reliability for each sensor in Eq. (2), it is more reasonable to weighted average the five sensors using the weights derived from their relative reliability (this process is a fundamental concept in data fusion study). In addition, the authors should clarify the justice of the magnitude of the error variance for each sensor. Did the authors setup a value arbitrarily or look up associated references? The resulting synthetic data may not be like the VODCA data which the authors are trying to simulate. Please classify.

Thank you for this comment. Our goal in this work was to showcase what can happen if measurement noise levels are not constant during the construction of multi-satellite data. Hence, we created our synthetic data in such a way as to illustrate potential pitfalls, and to see how well we could match global-scale patterns in vegetation resilience presented in previous work (e.g. Boulton et al., 2022; Smith et al., 2022). Our results are to some degree independent of the averaging scheme – whether data are merged via an average, weighted average, maximum value, or CDF-matching approach, underlying changes in noise levels will still be expressed. We chose to present a simplistic case (simple averages) with our synthetic data to make this point.

To address your concern, we have added a weighted-average version of our results as Figure 1 of this Reply. In essence, this is the same analysis as Figure 2 of the MS, but instead of taking an average for overlapping time periods, we take the weighted average, using weights defined by relative sensor reliability.

[Figure]

Figure 1: Replicate of Figure 2 from the MS using a weighted average – rather than simple average – to mix multiple sensors together.

As can be seen in Figure 1 of this Reply, the patterns we reveal do not alter – the effect we are trying to explore is not so much due to the averaging scheme chosen, but to the *fact of averaging disparate data itself* – changing the relative noise levels throughout the time series is what introduces biases, not the method of averaging data. There remain only slight variations in our overall statistics between the two averaging schemes.

Our quantitative choices for the noise levels (and averaging scheme) are thus not the main point of our study, and the chosen noise levels are less important in absolute terms than relative. We aimed to show how mixing data together can leave traces in resilience signals, which could be misinterpreted. We further aimed to match our 'jumps' in the resilience signals to the global-scale ones found in previous work (Smith et al., 2022). The similarity we find between aggregated synthetic and global VODCA indicates qualitatively that we construct synthetic data that illustrates the potential issues facing VODCA (or AVHRR).

In addition, why not create a bi-weekly synthetic data here to match the temporal resolution of the NDVI data? The temporal aggregation would add extra error into the final synthetic dataset.

We chose to first create daily data and then aggregate it to better match previous processing approaches. For example, AVHRR data is nominally daily data, but is time-aggregated to be more reliable against clouds and other errors. In previous work, VOD data has been similarly time-aggregated to match the temporal resolution of AVHRR data (Smith et al., 2022; Boulton et al., 2022). As we aimed to give insights that could be applied to real data, we chose to match the time-aggregation schemes of commonly used data sets, rather than construct temporally sparse data from scratch. In our tests, the

inferences were identical whether we applied our methods directly to the daily data or to the time-aggregated data; the main difference is the absolute – but not relative – values of variance and autocorrelation.

L. 82 Similar question with the VODCA simulation given that the authors are trying create a dataset to mimic the GIMMS3g NDVI.

Again, we chose to create daily data to mimic the underlying raw remote sensing data, and then time-aggregate following the methods used in previous papers.

L. 90 How did the authors to define the "changing land cover"? I understand that the authors are trying to get rid of anthropogenic effect. But all the land covers are affected by human. Please elaborate this sentence.

We defined changing land cover as any land cover that had changed classification over the period 2000-2020. For example, a pixel that had changed from Forest to Agriculture then back to Forest would be removed from our analysis, even if it was Forest at both the beginning and end of the study period. In this way we are conservative in only analyzing pixels with stable land cover. We have updated our description in the Methods.

L. 93 What resample method? The Maximum Value Composite or just averaging them?

We resample land cover to match the spatial resolution of the other data by taking the mode. We have clarified this in-text.

L. 107 The effects of combining multiple signals may further be muted if the authors considering weighted average (as I commented in L. 74).

[Figure]

Figure 2: Replicate of Figure 1 of the MS, with weighted rather than simple averaging.

There are indeed slight changes in resultant averaged time series when an alternative mixing scheme based on weighted averages is used. However, these differences are minor, and do not change the expression of dynamic noise levels on resilience indicators (Figure 1 of this Reply), which are sensitive to the relative strength of changing noise levels through time, not to the absolute values of those noise levels.

Caption of Fig.1. Why 0.44? Why not 0.3 or 0.2? Please clarify. This sounds like the authors have considered weighted average in the synthetic data construction. This confused me.

The reliability measures used here are arbitrary, and meant to mimic the global-scale patterns seen in the VOD data (cf. Figure 3). Our aim in this study was to see if we could replicate global-scale resilience signals (i.e., their time dynamics) in real multi-satellite data with a synthetic experiment; hence, noise values for each satellite are tuned towards that goal. We had not considered weighted averaging in our synthetic data construction, but have now added this as an additional check as Figure 1 of this Reply. While weighted averaging of course changes the *absolute* amount of noise variability, it does not change the *relative influence* of changing measurement noise through time (and hence the (anti)correlation between AC1 and variance). We maintain that the influence of time-variant noise levels will be expressed in multi-sensor data regardless of the data fusion scheme chosen. It is likely that a data fusion methodology could be created to mitigate this error, but the development of such a method is outside the scope of our study.

Fig. 2. Plot titles covered the words in (d), (f), and (h).

We are not sure which words you are referring to here – the plot titles do not cover any text. It is possible that you meant the x-axis label for (d), (f), (h)? Since those subplots all share an x-axis, we only labeled it on the bottom panels.

Fig. 2 and Fig. 3. Why use a five-year rolling window? Why not a longer or shorter window? Will the change of moving-time-window have an impact on your conclusions?

A five-year rolling window is fairly standard processing for time series of this length (Boulton et al. 2022; Smith et al. 2022) and since we investigate potential effects of the satellite composition on these previous works we decided to take the same window length. A longer or shorter window will not influence our results – the key point here is the time-variant noise levels. Whether or not those signals are found over shorter or longer windows, they will still influence inferred changes in resilience. We have added a version of Figure 2 of the MS with a 3- and 7-year rolling window to this Reply to illustrate this.

[Figure]

Figure 3: Same is MS Figure 3, but with a 3-year rolling window.

[Figure]

Figure 4: Same is MS Figure 3, but with a 7-year rolling window.

**References for this Reply**

Boers, Niklas. "Observation-based early-warning signals for a collapse of the Atlantic Meridional Overturning Circulation." Nature Climate Change 11.8 (2021): 680-688.

Boulton, C. A., Lenton, T. M. & Boers, N. Pronounced loss of amazon rainforest resilience since the early 2000s. Nature Climate Change 12, 271–278 (2022).

Feng, Y., Su, H., Tang, Z., Wang, S., Zhao, X., Zhang, H., Ji, C., Zhu, J., Xie, P., and Fang, J.: Reduced resilience of terrestrial ecosystems locally is not reflected on a global scale, Communications Earth & Environment, 2, 1–11, 2021.

Moesinger, L., Dorigo, W., de Jeu, R., van der Schalie, R., Scanlon, T., Teubner, I., and Forkel, M.: The global long-term microwave Vegetation Optical Depth Climate Archive (VODCA), Earth Syst. Sci. Data, 12, 177–196, https://doi.org/10.5194/essd-12-177-2020, 2020.

Pinzon, J. E. and Tucker, C. J.: A non-stationary 1981–2012 AVHRR NDVI3g time series, Remote sensing, 6, 6929–6960, 2014.

Smith, T., Traxl, D., and Boers, N.: Empirical evidence for recent global shifts in vegetation resilience, Nature Climate Change, 12, 477–484, 2022.

---

## Author Response (AR3)

**Reviewer 3**

Thanks for the authors great effort to address my concerns, especially the synthetic data construction. This significantly improved the robustness of this study. I only have a minor comment. Please note that the advance of current data fusion studies includes the effort to accommodate the second order moment in the fusion process by using the Triple Collocation Analysis (TCA). Nevertheless, I agree with the authors' argument that resilience/sensitivity metrics should take higher-order moment into consideration.

Thank you for taking the time with your review. We are delighted to hear that we could address your concerns with the Manuscript.

We have added a reference to the TCA method to our MS – indeed, that sounds like a very promising method to apply in our future analyses.